# Analyses of cancer incidence and other morbidities in neutron irradiated B6CF1 mice

**Alia Zander, Tatjana Paunesku, Gayle E. Woloschak**[ID]*

Feinberg School of Medicine, Radiation Oncology, Northwestern University, Chicago, Illinois, United States of America

* g-woloschak@northwestern.edu

**Data Availability Statement:** The data underlying the results presented in the study are available from http://janus.northwestern.edu/janus2/index.php.

## Abstract

The Department of Energy conduced ten large-scale neutron irradiation experiments at Argonne National Laboratory between 1972 and 1989. Using a new approach to utilize experimental controls to determine whether a cross comparison between experiments was appropriate, we amalgamated data on neutron exposures to discover that fractionation significantly improved overall survival. A more detailed investigation showed that fractionation only had a significant impact on the death hazard for animals that died from solid tumors, but did not significantly impact any other causes of death. Additionally, we compared the effects of sex, age first irradiated, and radiation fractionation on neutron irradiated mice versus cobalt 60 gamma irradiated mice and found that solid tumors were the most common cause of death in neutron irradiated mice, while lymphomas were the dominant cause of death in gamma irradiated mice. Most animals in this study were irradiated before 150 days of age but a subset of mice was first exposed to gamma or neutron irradiation over 500 days of age. Advanced age played a significant role in decreasing the death hazard for neutron irradiated mice, but not for gamma irradiated mice. Mice that were 500 days old before their first exposures to neutrons began dying later than both sham irradiated or gamma irradiated mice.

## Introduction

Ionizing radiation can be classified by its linear energy transfer (LET) to better understand how quickly radiation is attenuated and the concentration of energy deposited near the particle track. Experiments evaluating the biological effects of low and high LET ionizing radiation found that high LET radiation is more damaging to biological material in part because sites of DNA double strand damage and other types of damage are in closer proximity to one another than occurs with low LET radiation. These clusters of damage make DNA repair more difficult and lead to increased cell death [1]. Neutrons are the most penetrating of all particles included in the category of high LET ionizing radiation. Neutrons are further classified by the amount of kinetic energy in a free neutron and this energy ranges from less than 0.02eV for cold neutrons to over 20MeV for ultrafast neutrons. While numerous animal studies were conducted

**Funding:** National Institute of Health grants R01OH010469 and RO1CA221150 were both awarded to GEW.

**Competing interests:** The authors declare no competing interests. The corresponding author of this paper, Gayle E. Woloschak, is a section editor for PLOS ONE. This does not alter our adherence to PLOS ONE policies on sharing data and materials.

in the USA, Europe and Asia using low LET radiation such as x-rays and gamma rays [2–5], extensive neutron irradiation experiments were less common.

Between 1972 and 1989, Argonne National Laboratory (ANL) housed one of the few neutron irradiators called the JANUS reactor suitable for large-scale whole-body animal irradiations with fission spectrum neutrons with an energy range peak at 1 MeV [6–10]. During this time, ANL performed ten independent experiments investigating the effects of gamma and neutron irradiations on mice [11]. These animals were irradiated under different conditions and allowed to live out their entire lifespan. Ionizing radiation exposures ranged from low to high doses, included fractionated and acute exposures, and the age when mice were first exposed to ionizing radiation varied by experiment. Moribund mice were necropsied to determine a cause of death (COD), other death contributing diseases, and non-contributing diseases. The resulting dataset from the Janus studies includes information on over 50,000 male and female mice [3, 11–13]. A neutron irradiation study of this magnitude will most likely never be reproduced and it is important that researchers continue to analyze these data in light of novel biological findings and analytical methods. Innovative statistical approaches, such as the ones used here, can lead to new insights into the effects of neutron irradiation on whole organisms.

In this study we analyzed neutron irradiated mice using methods similar to those used for investigation of cobalt 60 gamma irradiated mice [14]. Here, we included comparisons between key findings from our analyses on neutron and gamma irradiated mice. Comparisons between the two qualities of radiation are beneficial for determining the underlying biological factors leading to differential health outcomes. Numerous *in vivo* studies comparing x-rays or gamma rays with neutrons evaluated life shortening or focused on genetic changes to establish a suitable relative biological effectiveness factor (RBE) [6–9, 15–21]. Work with radioprotectors and neutron irradiation *in vivo* and *in vitro* also found profound differences between the high and low LET ionizing radiation [22–25].

Previous neutron irradiation studies suggested that fractionation had little effect on overall survival or cancer risk from neutron exposures [6, 7, 9, 26–31]. We re-analyzed data from the Janus experiments conducted at ANL by pooling results from different studies in a new way in order to determine the role of fractionation in neutron irradiations for overall survival and for risks of specific causes of death. Rigorous statistical testing on sham irradiated mice enabled us to combine several Janus experiments into one large dataset. This approach increased our statistical power and allowed comparisons of different fractionation regimens across a wide array of total doses [14]. A more voluminous dataset generated by this approach allowed us to explore neutron fractionation effects in young and old mice across a wide array of total doses of fission spectrum neutrons.

## Methods

### Data selection

Details on data selection methods used in this study can be found in our previously published work [14] and **S1 Table in** S1 File. To increase the statistical power given the limited set of fractionation schedules for neutron irradiated mice, we combined mice that received their total doses in 24 and 60 fractions into a single group. Therefore, in addition to the initial analysis of control mice, we also verified that grouping together 24 and 60 sham irradiation fractions did not significantly change survival probabilities in control mice compared to animals that received acute sham exposures. Because the greatest number of fractions a neutron irradiated mouse received was only 60 fractions, we also excluded all animals with more than 60 sham irradiation fractions from this work.

We used Cox proportional hazards models with sex and a categorical fractionation term as independent variables for the main model (**S1A and S1B Fig in** S1 File). For sensitivity analysis, we also stratified by sex (**S1C and S1D Fig in** S1 File) and found no significant changes in the model output. In addition, we tested the impact of using a mixed-effects model to account for variability between experiments. The results in **S2 Table in** S1 File show similar results to the main model. Model output showed that the integrated and penalized log likelihoods were very similar and both significant (p-value < 0.001), increasing our confidence that the control experiments outlined in our previous paper [14] were sufficient for capturing variability between experiments. Finally, we used Kaplan Meier survival curves to validate the proportional hazards assumption in our model and found non-overlapping survival curves between sham irradiated mice that received acute and fractionated exposures (**S1E Fig in** S1 File). After filtering out any control mice that received more than 60 sham irradiation fractions, we again verified that there were no significant differences between control mice based on experiment number or age first irradiated. **S1 Table in** S1 File outlines the filtering process used to determine an acceptable cohort of neutron irradiated mice that combined data from multiple Janus experiments.

## Survival analysis

Kaplan-Meier (KM) curves were used for categorical univariate survival analysis in R through the survfit function within the survival package [32, 33]. Cox proportional hazard (PH) models were utilized to evaluate survival over time for multivariate models. Cox PH models enabled us to include interactions between variables, incorporate quantitative and categorical variables, and stratify by variables that do not have proportional hazards [34]. The main Cox PH model used for neutron irradiated mice was as follows:

$$\lambda(t) = \lambda_0(t)e^{(\beta_1 sex + \beta_2 first\ irrad + \beta_3 total\ dose +\ \beta_4 fractionated + \beta_5 total\ dose:fractionated)}$$

where $\lambda(t)$ is the hazard function based on our set of covariates (sex, age first irradiated, total dose, fractionated (acute vs. fractionated), and the interaction between total dose and fractionated), $\boldsymbol{\beta}$ is a vector of their associated parameter estimates, and $\lambda_0(t)$ is the baseline hazard. All Cox PH models were created in R through the coxph function from the survival package [33]. We checked for multicollinearity in our models using the vif function from the car package in R. As expected, there was multicollinearity with the interaction term, however this was not an issue in our model because the standard error in our models was not overly inflated. Additionally, sensitivity analyses showed consistent results across different models.

## Competing risks analysis

We conducted competing risks analyses to understand the risks and probabilities of specific causes of death. In situations where there are no competing risks, or events that could decrease the likelihood of observing the event of interest (i.e. other causes of death), the cumulative incidence of an event is calculated as the one minus the survival function estimated with KM curves. However, when competing risks are present, the KM method is biased upward and the Fine and Gray method for CIF is more appropriate for examining probabilities of particular event types [35]. For our competing risks analyses, we assessed crude incidences, cause-specific hazard models, and cumulative incidence function (CIF) regression models (also known as the subdistribution hazard) [36]. To investigate crude, nonparametric incidences with competing risks present, we utilized the cuminc function from the cmprsk package in R [37].

We utilized both cause specific hazards and CIF models for multivariate regression analyses in the presence of competing risks. Cause specific hazards are used to determine the effect that

covariates have on all event free subjects. These hazards were estimated using the coxph function from the survival package in R [33]. For this method all causes of death other than the event of interest, were censored. Concretely:

$$\lambda_k(t) = \lambda_{0k}(t)e^{(\beta_1 sex + \beta_2 first\ irrad + \beta_3 total\ dose + \beta_4 fractionated + \beta_5 total\ dose:fractionated)},$$

subset data for age first irradiated < 500 days.
where $\lambda_k(t)$ is the hazard function for the k[th] cause of death.

The cumulative incidence function describes the overall likelihood of a specific result and does not depend on a subject being event free [35, 38–40]. We utilized the crr function in the cmprsk package in R to model CIF regression hazards [37]. Concretely:

$$\lambda_k^*(t) = \lambda_{0k}(t)e^{(\beta_1 sex + \beta_2 firsr\ irrad + \beta_3 total\ dose + \beta_4 fractionated + \beta_5 total\ dose:fractionated)},$$

subset data for age first irradiated < 500 days.
where $\lambda_k^*(t)$ is the subdistribution hazard function for the k[th] cause of death.

## Cause of death animal groupings

The full list of causes of death separated by radiation treatment conditions (control, gamma rays, neutrons) and subdivided by sex is shown in **S3 Table in** S1 File. "Grouped Macros" data was downloaded from the Janus website and includes all pathologies found at necropsy. CODs were categorized as lethal (L), contributory (C), or non-contributory (N) [11]. For the purposes of this study, we only investigated lethal disorders. We grouped causes of death as solid tumors other than lymphomas (referred to as tumors), lymphomas, non-tumors, or cause of death unknown (CDU) to gain a better understanding of disease trends. For certain analyses, we also looked at lung tumors separately from all other non-lymphoma solid tumors and examined non-thymic lymphoma specifically.

## Survival analysis for JM8 mice

One of the Janus experiments excluded from the main analysis and treated separately is the experiment JM8 where animals received once a week a 45 minutes sham fraction or a fraction of gamma rays (dose rates of 0.15, 0.37, or 0.68 cGy/min), or neutrons (dose rates 0.014, 0.035 or 0.056 cGy/min). Because irradiations continued until the death of the animals, it was impossible to ascribe discrete total doses in this study.

The main Cox PH model used for gamma irradiated JM8 mice analysis was:

$$\lambda(t) = \lambda_0(t)e^{(\beta_1 sex + \beta_2 dose\ rate\ (0.148) + \beta_3 dose\ rate\ (0.370) + \beta_4 dose\ rate\ (0.680))},$$

all mice included.

The main Cox PH model used for neutron irradiated JM8 mice analysis was:

$$\lambda(t) = \lambda_0(t)e^{(\beta_1 sex + \beta_2 dose\ rate\ (0.0138) + \beta_3 dose\ rate\ (0.0349) + \beta_4 dose\ rate\ (0.0558))},$$

all mice included.

## Results

### Fractionation decreases the death hazard in mice exposed to neutrons

We examined age at death in mice from Janus experiments for different total doses and fractionation regimens (Fig 1). The range of total doses for neutron irradiated mice was 0.94cGy - 301cGy. A minor subset of animals was first exposed to neutrons when they were over 500

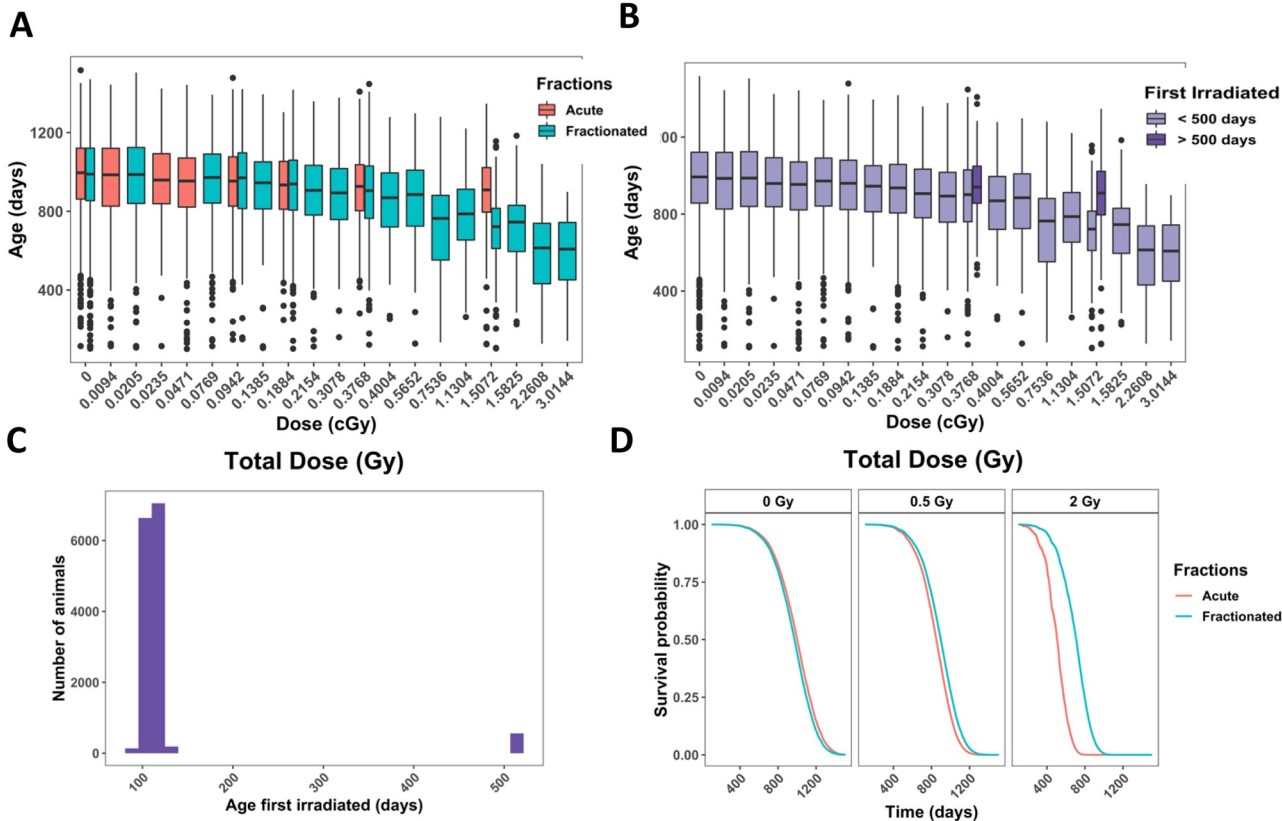

**Fig 1. Analysis on animals filtered in S1 Table in S1 File including controls or neutron irradiated.** (A) Box plot of age at death in days versus total dose in Gy. Colors indicate if the exposure was acute or fractionated. (B) Age at death in days versus total dose in Gy. Colors indicate whether a mouse was first irradiated before (light) or after (dark purple) 500 days. (C) Histogram of the total number of animals versus age first irradiated in days. (D) Representative graphs from Cox PH model output with age at death as the time scale and sex, age first irradiated, total dose, fractions, and the interaction between total dose and fractions as independent variables. The predicted outcomes shown are for female mice first irradiated at 120 days. All independent variables were significant in the model, as shown by the parameter estimates and statistical output from the model (Table 1).

days old, while the remainder of the mice were first irradiated around 100 days +/- 15 days (Fig 1B). The highest acute exposure neutron dose was 37.68cGy for young mice and 150.72cGy for aged mice. When comparing the age at death for aged mice compared to mice first irradiated around 100 days, there was a noticeable increase in longevity in the aged irradiated mice. No equivalent finding was noted in the gamma irradiated young and old mice cohorts (Fig 1C and [14]). We excluded the aged irradiated mice from our main analysis because none of the sham irradiated animals in our cohort were aged before sham exposure and this small subset of data would have had a large amount of leverage on our model. Aged mice were, however, incorporated in robustness tests. Excluding these 560 aged mice from further investigation resulted in 14,018 mice total for our neutron analysis.

The relative biological effectiveness (RBE) used to set up gamma ray and neutron irradiations was based on life shortening per cumulative dose; it was estimated to be around 10 for neutron doses up to 40cGy and approximately 5 or less for neutron doses over 40cGy ([11]– page 36). Because of this RBE estimate, the maximal total dose for neutrons was approximately one order of magnitude lower than the maximum dose used for gamma irradiations. In addition, the fractionation schedule was more limited and neutron irradiated mice only received 24 or 60 fractions. We combined these two fractionation conditions to increase sample size.

**Table 1. Parameter estimates, hazard ratios with 95% confidence interval, and p-values for main Cox PH model.**

| term | Estimate | Hazard Ratio (95% CI) | P-value |
|---|---|---|---|
| Sex(M) | -0.19 | 0.828 (0.790, 0.868) | <**0.001** |
| Fractionated | 0.165 | 1.179 (1.105, 1.259) | <**0.001** |
| Total Dose | 1.85 | 6.36 (4.13, 9.78) | <**0.001** |
| Age First Irradiated | -0.003 | 0.997 (0.9943, 0.9998) | **0.040** |
| Fractionated:Total Dose | -0.933 | 0.39 (0.26, 0.61) | <**0.001** |

Using Cox PH models to study the effects of fractionation on survival in neutron irradiated mice, we found that the key interaction term between fractionations and total dose significantly decreased the death hazard (Fig 1D and Table 1). Similar to gamma irradiated mice, males had a significantly lower death hazard compared to female mice and the main effects for fractionation and total dose significantly increased the hazard. Even with a limited range of ages from 90 to 180 days, an increase in age first irradiated significantly decreased the hazard. All of these variables showed statistical significance in our model, including the age first irradiated, a variable that was not significant in gamma irradiated mice [14]. Robustness tests showed similar results, increasing our confidence in the benefit of fractionation on neutron irradiated mice (**S2 Fig in** S1 File). When we included aged mice for our robustness testing, acute exposures appeared to be less hazardous compared to fractionated exposures (**S2A and S2B Fig in** S1 File). However, after correcting the model by adding in an interaction term between age first irradiated and total dose to account for the increased rescue effect of older ages at higher doses (Fig 1C), fractionation decreased the death hazard (**S2C and S2D Fig in** S1 File). Adding an interaction term between age first irradiated and total dose for the main model did not change the significance nor the sign of any terms in the model, and the interaction term itself was not significant (**S2E and S2F Fig in** S1 File). Stratifying by sex (**S2G and S2H Fig in** S1 File) and including exact fractionation regimen and age first irradiated for control mice (**S2I and S2J Fig in** S1 File) had no major consequence on the model. We used KM curves to validate the proportional hazards assumption in our model and found parallel survival curves between groups based on sex, number of fractions, age first irradiated, and total dose (**S3 Fig in** S1 File).

## In neutron irradiated mice, fractionation significantly decreases the hazard for tumor deaths

Accounting for competing risks by calculating cause-specific hazards, we found that the interaction between total dose and the fractionation variable was significant for tumor deaths (Fig 2A and Table 2) where fractionation was beneficial for survival. In animals that developed lymphoma (Fig 2B and Table 2), non-tumors (Fig 2C and Table 2) or CDU (Fig 2B and Table 2) fractionation lead to differences that were not significant. In Fig 2B and 2C the line representing acute exposures is hidden directly behind the line showing fractionated exposures. To evaluate this further, we inspected death due to lung tumors (**S4A Fig in** S1 File), non-thymic lymphoma (**S4C Fig in** S1 File), and tumors other than lung tumors (**S4B Fig in** S1 File). The interaction term between fractionation and total dose was not significant for lung tumors nor non-thymic lymphoma, but was still significant in tumors other than lung tumors (**S4B and S4D Fig in** S1 File and Table 2). Males had a lower death for all specific causes of death, except for tumors. Males had a higher hazard for lung tumors specifically, but not for tumors excluding lung tumors, indicating that lung tumor risk in males was driving the entire outcome for the tumor group.

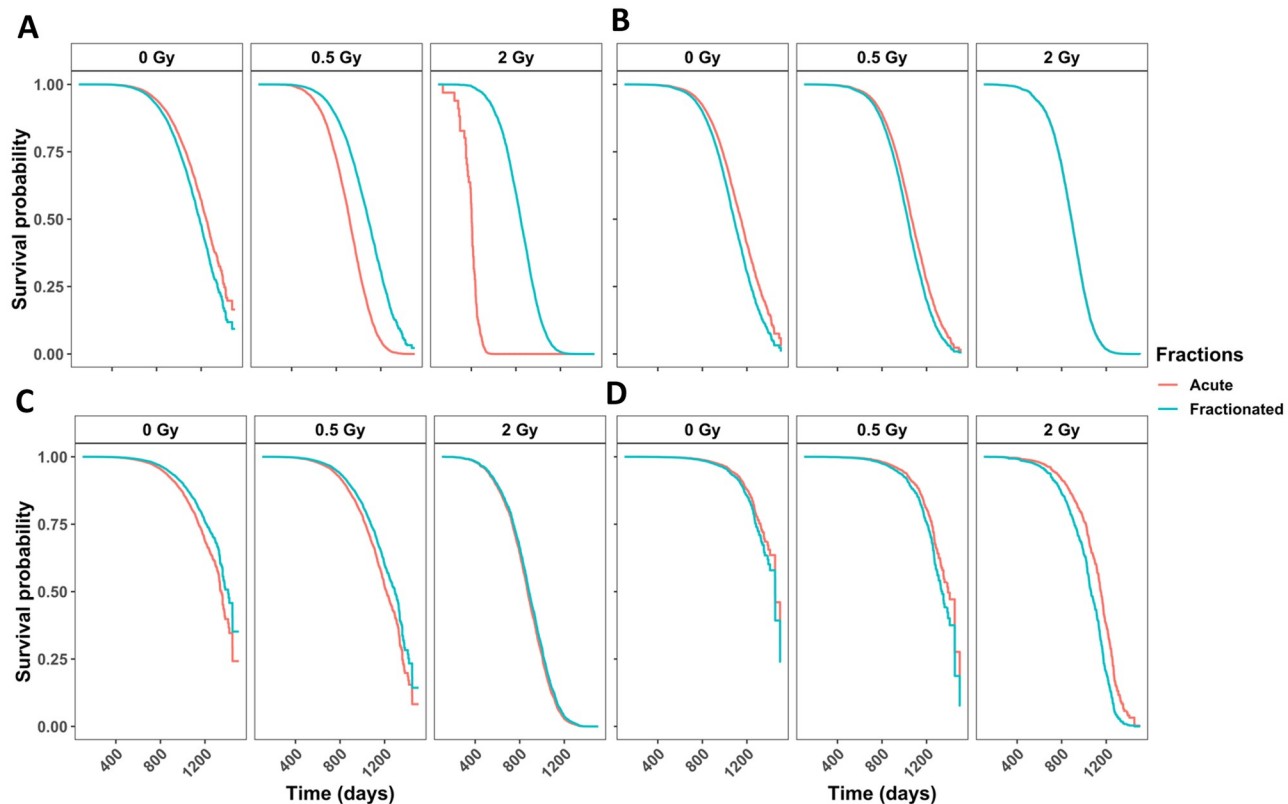

**Fig 2. Competing risks models for specific causes of death in neutron-irradiated mice with age as a time scale and sex, age first irradiated, total dose, fractions, and the interaction between total dose and fractions as independent variables.** Survival curves for causes of death being (**A**) any solid tumors, (**B**) lymphoma, (**C**) non-tumors, and (**D**) cause of death unknown. Model estimates, confidence intervals and p-values are listed in Table 2. The graphs represent predicted outcomes for female mice first irradiated at 120 days.

### Lymphoma is the most common COD in neutron irradiated mice while sex significantly impacts probabilities for most COD in control and neutron irradiated mice

Fig 3A shows the non-parametric cumulative incidence for each main COD. Lymphoma was the most prevalent, followed by lung tumors, all remaining tumors, non-tumors, and CDU. After dividing the data into control (Fig 3B) and neutron-irradiated mice (Fig 3C), we observed that the earliest cases of death occurred around 750 days in both groups of animals. These graphs were also showing sex differences for specific causes of death. For control and neutron-irradiated mice, males had a lower incidence of lymphomas, tumors excluding lung tumors, and non-tumors, but females had a much lower incidence of lung tumors. The difference between males and female mice was significant for both controls and neutron irradiated mice for all causes of death, except CDU (Fig 3B and 3C).

### Tumors other than lung tumors were more frequent with acute neutron exposures

It is important to examine subdistribution hazards, also known as cumulative incidence functions in addition to cause-specific hazards [35, 36, 38–40]. The parameter estimates have a less direct interpretation using the Fine and Grey method for CIFs, but explain the overall likelihood of a specific outcome. We included sex, age first irradiated, number of fractions, total

**Table 2. Competing risks model output for cause specific hazards and subdistribution hazards.**

| | | Cause Specific Hazards | | | Subdistribution Hazards | | |
|---|---|---|---|---|---|---|---|
| **COD** | **Independent Variable** | **Estimate** | **Hazard Ratio (95% CI)** | **P-value** | **Estimate** | **Hazard Ratio (95% CI)** | **P-value** |
| Tumors | sexM | 0.17 | 1.185 (1.101, 1.276) | **<0.001** | 0.482 | 1.62 (1.508, 1.74) | **<0.001** |
| | Fractions | 0.275 | 1.32 (1.18, 1.47) | **<0.001** | 0.027 | 1.027 (0.941, 1.121) | 0.55 |
| | Total dose | 3.362 | 28.8 (15.1 54.9) | **<0.001** | 0.199 | 1.221 (0.974, 1.53) | 0.083 |
| | First irrad | -0.008 | 0.993 (0.987 0.998) | **0.005** | 0 | 1 (0.999, 1.001) | 0.88 |
| | Fractions: Total dose | -2.421 | 0.09 (0.05, 0.17) | **<0.001** | -0.023 | 0.977 (0.773, 1.235) | 0.85 |
| Lymphoma | sexM | -0.46 | 0.634 (0.587, 0.685) | **<0.001** | -0.38 | 0.69 (0.64, 0.74) | **0** |
| | Fractions | 0.29 | 1.33 (1.20, 1.47) | **<0.001** | 0.22 | 1.25 (1.15, 1.36) | **<0.001** |
| | Total dose | 0.749 | 2.12 (1.01, 4.43) | **0.047** | -0.429 | 0.651 (0.499, 0.85) | **0.002** |
| | First irrad | 0.003 | 1.003 (0.997, 1.008) | 0.310 | 2.75E-04 | 1.000 (1.000, 1.001) | 0.46 |
| | Fractions: Total dose | -0.138 | 0.87 (0.42, 1.83) | 0.714 | 0.026 | 1.027 (0.776, 1.359) | 0.85 |
| Non-tumors | sexM | -0.49 | 0.61 (0.42, 0.69) | **<0.001** | -0.34 | 0.71 (0.63, 0.80) | **<0.001** |
| | Fractions | -0.306 | 0.74 (0.63, 0.86) | **<0.001** | -0.40 | 0.67 (0.58, 0.77) | **<0.001** |
| | Total dose | 1.128 | 3.09 (1.06, 8.99) | **0.039** | 0.445 | 1.56 (1.119, 2.175) | **0.009** |
| | First irrad | -0.009 | 0.991 (0.98, 0.999) | **0.016** | 0 | 1 (0.999, 1.001) | 0.62 |
| | Fractions: Total dose | 0.114 | 1.12 (0.38, 3.26) | 0.835 | 0.144 | 1.155 (0.819, 1.627) | 0.41 |
| CDU | sexM | -0.30 | 0.74 (0.62, 0.89) | **0.002** | -0.13 | 0.883 (0.74, 1.05) | 0.17 |
| | Fractions | 0.19 | 1.21 (0.93, 1.57) | 0.163 | 0.023 | 1.024 (0.824, 1.272) | 0.83 |
| | Total dose | 1.011 | 2.75 (0.38, 20.05) | 0.318 | -0.019 | 0.981 (0.568, 1.694) | 0.95 |
| | First irrad | 0.003 | 1.003 (0.991, 1.015) | 0.659 | 0.001 | 1.001 (0.999, 1.002) | 0.53 |
| | Fractions: Total dose | 0.157 | 1.17 (0.16, 8.54) | 0.877 | 0.303 | 1.354 (0.771, 2.378) | 0.29 |
| Lung tumors | sexM | 0.79 | 2.20 (1.97, 2.46) | **<0.001** | 1.12 | 3.08 (2.76, 3.43) | **<0.001** |
| | Fractions | 0.206 | 1.23 (1.02, 1.47) | **0.026** | 0.13 | 1.14 (0.99, 1.32) | 0.071 |
| | Total dose | 1.078 | 2.94 (0.71, 12.12) | 0.136 | 0.03 | 1.03 (0.784, 1.353) | 0.830 |
| | First irrad | -0.011 | 0.989 (0.982, 0.997) | **0.008** | 0.001 | 1.00 (1.000, 1.002) | **0.027** |
| | Fractions: Total dose | -0.254 | 0.78 (0.19, 3.20) | 0.726 | -0.006 | 0.994 (0.748, 1.323) | 0.97 |
| Tumors (excluding lung tumors) | sexM | -0.44 | 0.643 (0.58, 0.72) | **<0.001** | -0.268 | 0.765 (0.691, 0.847) | **<0.001** |
| | Fractions | 0.263 | 1.30 (1.13, 1.50) | **<0.001** | -0.065 | 0.937 (0.835, 1.052) | 0.27 |
| | Total dose | 4.218 | 67.9 (32.9, 140.3) | **<0.001** | 0.526 | 1.691 (1.067, 2.681) | **0.025** |
| | First irrad | -0.006 | 0.994 (0.987, 1.001) | 0.093 | -0.002 | 0.998 (0.997, 1) | **0.024** |
| | Fractions: Total dose | -3.169 | 0.04 (0.02, 0.09) | **<0.001** | -0.312 | 0.732 (0.458, 1.17) | 0.19 |

dose, and the interaction between fractions and total dose as independent variables in our subdistribution hazards model. The competing risk groups included lymphomas, lung tumors, tumors excluding lung tumors, non-tumors, and CDU. We discovered that females were more susceptible to tumors excluding lung tumors. Specifically, the incidence of tumors excluding lung tumors increased with increase of total dose and decreased with fractionation (Table 2). By graphing predicted outcomes under varying conditions, we found that the model predicted that when there was a tenfold difference in total dose, fractionation was the largest determinant for tumor incidence, with acute exposures resulting in the most tumors (Fig 3D). When model was generated for the hundredfold difference in total dose, fractionation remained the dominant factor modulating outcome. Acute high dose exposures resulted in the most tumor incidences in all scenarios (Fig 3E).

Examining lung tumors specifically, we found that males were more prone to lung tumors and increases in total dose and fractionation contributed to increased lung tumor incidence although not significantly (Table 2). Plotting predicted outcomes under variable conditions for the tenfold difference between high and low total doses, fractionation was the major indicator

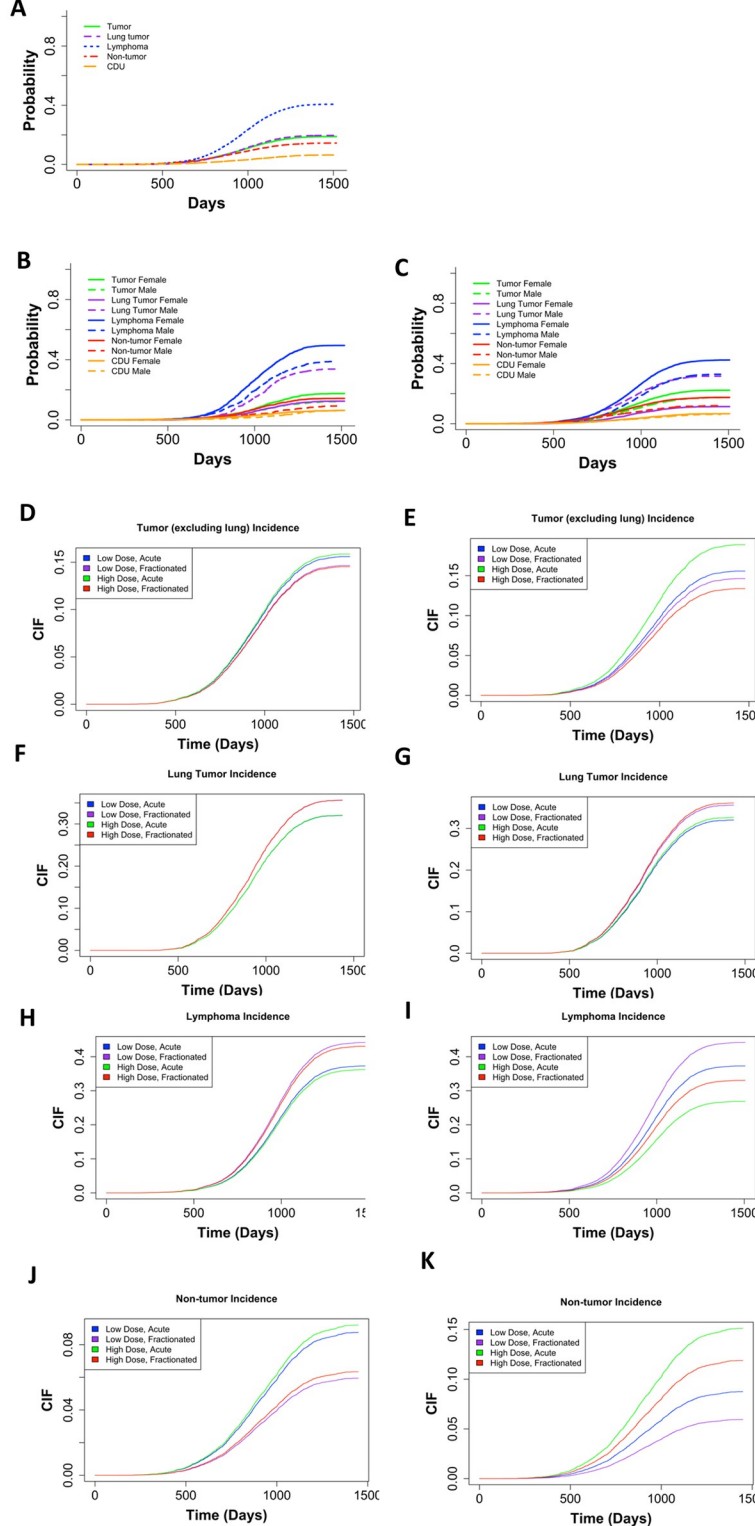

**Fig 3.** (**A**) Non-parametric CIF for the five main categories of COD without grouping. (**B**) Non-parametric CIF for the five main categories of COD grouped by sex for control/sham irradiated mice or (**C**) neutron-irradiated mice. P-values for differences in sex for each COD for mice graphed in (**B**): Tumor = 1.11E-05, lung tumor = 0, lymphoma = 3.75E-11, non-tumor = 2.01E-06, CDU = 0.66, and for mice graphed in (**C**): tumor = 1.783E-08, lung tumor = 0, lymphoma = 0, non-tumor = 7.07E-12, CDU = .453. Predicted outcome under the following conditions:

low dose = 0.01Gy, high dose = 0.1Gy for (**D**) tumors (excluding lung), (**F**) lung tumors, (**H**) lymphomas, and (**J**) non-tumors. Predicted outcome under the following conditions: low dose = 0.01Gy, high dose = 1Gy for (**E**) tumors (excluding lung), (**G**) lung tumors, (**I**) lymphomas, and (**K**) non-tumors. All predicted outputs model males first irradiated at 120 days. Model output with parameter estimates, hazard ratios (95% confidence interval), and p-values are listed in Table 2.

for lung tumor incidence (Fig 3F). When the difference between high and low total doses was hundredfold, fractionation again correlated with the greater lung tumor occurrences (Fig 3G).

## Lymphoma deaths were most frequent in mice that received fractionated low-dose-range neutron exposures

When examining lymphoma deaths, we found that females were at higher risk than males. Increasing the total dose of neutrons decreased lymphoma death incidence, fractionation increased it (Table 2). Predicted outcome graphs showed that when the difference between high and low total doses was tenfold, fractionation was the most important determinate for increased lymphoma incidence (Fig 3H). Conversely, when the difference between high and low total doses was hundredfold, dose became the leading factor and lower total dose exposures resulted in the most lymphoma incidences (Fig 3I). Under all conditions, low dose fractionated exposures resulted in the greatest number of lymphoma incidences.

## Non-tumor deaths were most common after acute high dose exposures

Non-tumor deaths were more prevalent in female mice compared to male mice, increasing with an increase in total dose. Fractionation decreased the probability of death due to non-tumors (Table 2). In the graph of predicted outcomes for the tenfold difference between high and low total doses, either acute exposure results in the most non-tumor cases (Fig 3J). Conversely, when the difference between high and low total doses was hundredfold, dose became the dominant factor and high-dose conditions result in the most non-tumor death incidences (Fig 3K). Under all conditions, high dose acute exposures resulted in the greatest non-tumor incidence.

## A 60cGy cut-off does not impact CIF outcomes in neutron irradiated mice

Unlike in gamma ray irradiated mice [14], no shoulder was observed in the lymphoma or non-tumor incidence graphs for neutron irradiation. Removal of mice exposed to neutrons with total doses of 60cGy or greater did not change the CIF output graphs as it did after removal of mice exposed to high doses of gamma rays (**S5 Fig in** S1 File).

## Mice first exposed to neutrons after 500 days of age began dying later and had increased incidences of death due to tumors

Because of the pronounced difference in longevity of aged mice exposed to neutrons, we examined causes of death in mice irradiated at 500 days of age versus those that were irradiated younger. In addition, we compared old neutron irradiated animals with the mice exposed to gamma rays at 500 days of age. Control mice (Fig 4A) and gamma irradiated mice first irradiated before 200 days of age (Fig 4B and 4C) all displayed similar trends in COD patterns with lymphomas as the most prevalent, followed by tumors, non-tumors, and CDU. Control mice (Fig 4A), aged gamma-irradiated mice (Fig 4D), and mice that were first gamma irradiated at a young age began dying around 600 days. However, data graphs for mice exposed to 6 Gy or more of gamma rays showed a shoulder with increased death incidence beginning around 300

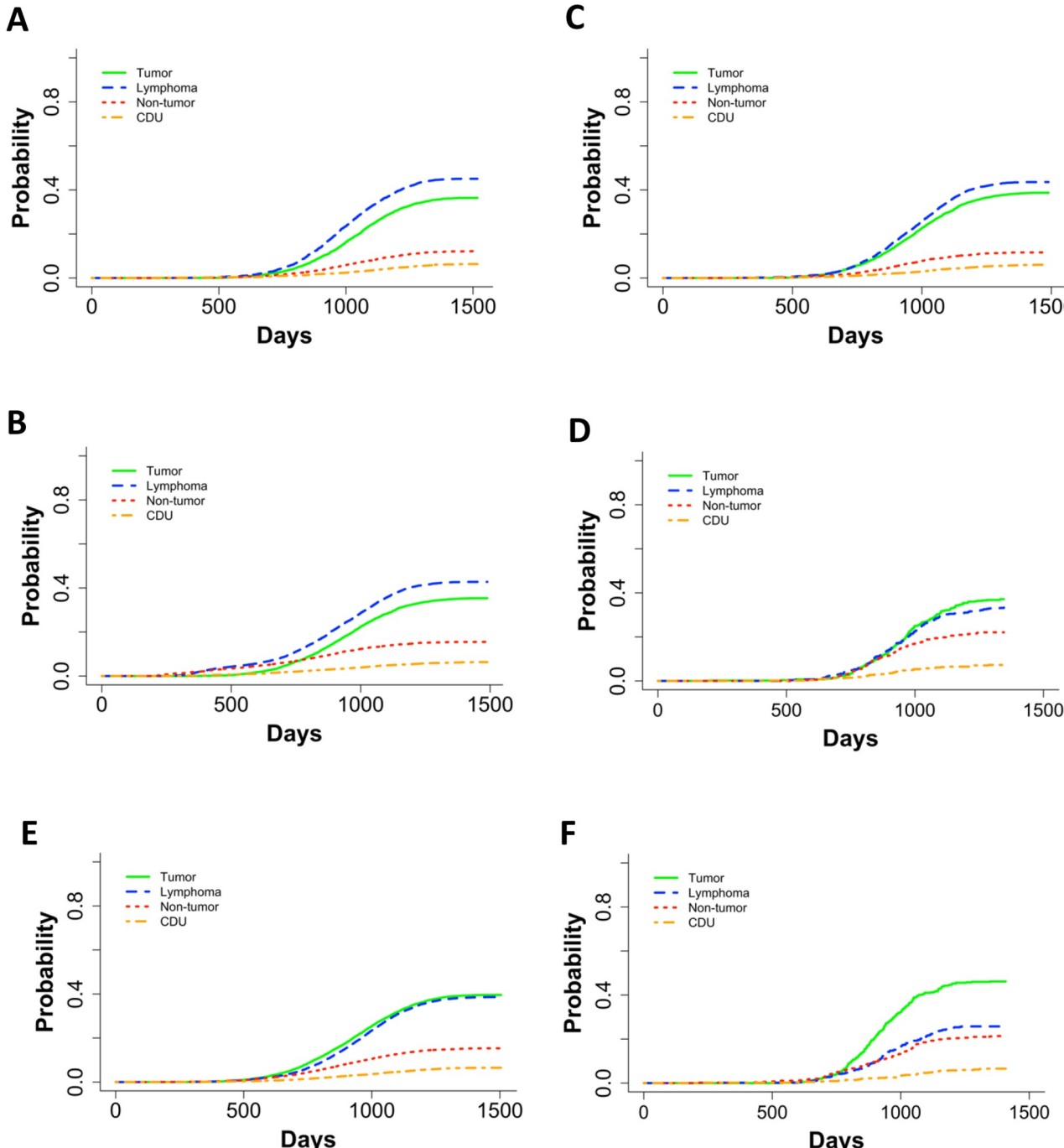

**Fig 4.** Non-parametric CIFs for the four main categories for COD in **(A)** control mice, **(B)** gamma irradiated mice first irradiated under 200 days old, **(C)** gamma irradiated mice that were first irradiated under 200 days old with a dose cut-off of 6Gy, **(D)** gamma irradiated mice first irradiated at or after 500 days, **(E)** neutron irradiated mice first irradiated under 200 days old, and **(F)** neutron irradiated mice first irradiated over 500 days old. P-values for differences between groups of ages first irradiated in **C and D**: tumor = 0.73, lymphoma = 7.03E-05, non-tumor = 9.73E-13, CDU = 0.09, and in **E and F**: tumor = 2.30E-03, lymphoma = 3.71E-08, non-tumor = 2.49E-04, and CDU = 0.97.

days of age associated with lymphomas and non-tumors (Fig 4B). In gamma irradiated aged mice (Fig 4B), there were significantly more cases of non-tumors and significantly fewer cases of lymphoma in comparison to mice exposed at a young age. In neutron-irradiated mice, solid tumors were as frequent as lymphomas when they were irradiated at a young age (Fig 4E).

This became more evident in neutron irradiated aged mice (Fig 4F) where tumors were markedly more frequent than lymphomas. Aged mice died significantly more often with tumors and non-tumors compared to younger neutron irradiated mice, and there were significantly fewer deaths from lymphomas in aged mice (Fig 4). Mice first irradiated with neutrons at age of 500 days did not begin dying until close to 750 days of age, while mice first irradiated at age close to 115 days began dying around 600 days. Thus, aged neutron-irradiated mice began dying later in life than the control mice (Fig 4F compared to Fig 4A).

### Gamma and neutron irradiated mice exposed to fractionated once weekly irradiation had a significantly greater death hazard with increase in weekly dose/dose rate

Janus experiments JM8 included mice that were irradiated once a week until death for 45 minutes at a time with dose rates of 0.15, 0.37, or 0.68 cGy/min with cobalt 60 gamma rays or 0.014, 0.035 or 0.056 cGy/min with fission spectrum neutrons. Because the total dose in this experiment was dependent on the lifespan, we performed a separate analysis with Cox PH models using dose rate as our independent variable. We found that as the dose rate increased, the death hazard increased significantly for mice exposed to gamma rays or neutrons (Fig 5A and 5B, and **S4 Table in** S1 File). Sensitivity analyses for gamma- and neutron-irradiated mice (**S5 and S6 Tables in** S1 File) supported these results. We examined KM curves to validate the proportional hazards assumption for using CoxPH models (**S6 Fig in** S1 File). The KM curves for sex in neutron-irradiated mice revealed that male and female survival curves overlapped (**S6D Fig in** S1 File). However, when we tested for an interdependence between residuals and time using Schoenfeld residuals (**S6E Fig in** S1 File), we found a non-significant relationship, therefore validating our use of Cox PH models.

### Sex variable is significant in weekly exposed neutron irradiated but not gamma irradiated JM8 mice

In the JM8 experiment, sex played a significant role in neutron-irradiated mice, but not gamma-irradiated mice (**S4 Table in** S1 File). The JM8 dataset consisted of 392 male and 215 female neutron exposed mice and 395 male and 215 female mice gamma irradiated animals. This indicates that sample size is not contributing to the difference in significance for the sex term in our Cox PH model utilizing gamma irradiated mice. Upon further investigation through non-parametric CIFs, we found that control mice only exhibited a significant difference between sexes for lung tumors (Fig 5C). Neutron irradiated mice had a significant difference between sexes for lung tumors, tumors (excluding lung tumors), and lymphomas (Fig 5D). Gamma irradiated mice only displayed a significant difference between lung tumors and non-tumors (Fig 5E). For all treatment groups, females were at a greater risk than male mice for every COD except lung tumors.

## Discussion

Age first irradiated played a significant role in decreasing the death hazard in neutron-irradiated mice. This was evident by examining box plots and the main CoxPH model output for neutron irradiated mice. Age first irradiated was not a significant variable in the main model for gamma irradiated mice [14], but it was a significant term in the main model output for neutron-irradiated mice (Table 1). A significant interaction between age first irradiated and total dose for neutron irradiated mice was discernible when aged mice data were added to the analysis. As dose increased, the importance of age first irradiated also increased. Among the

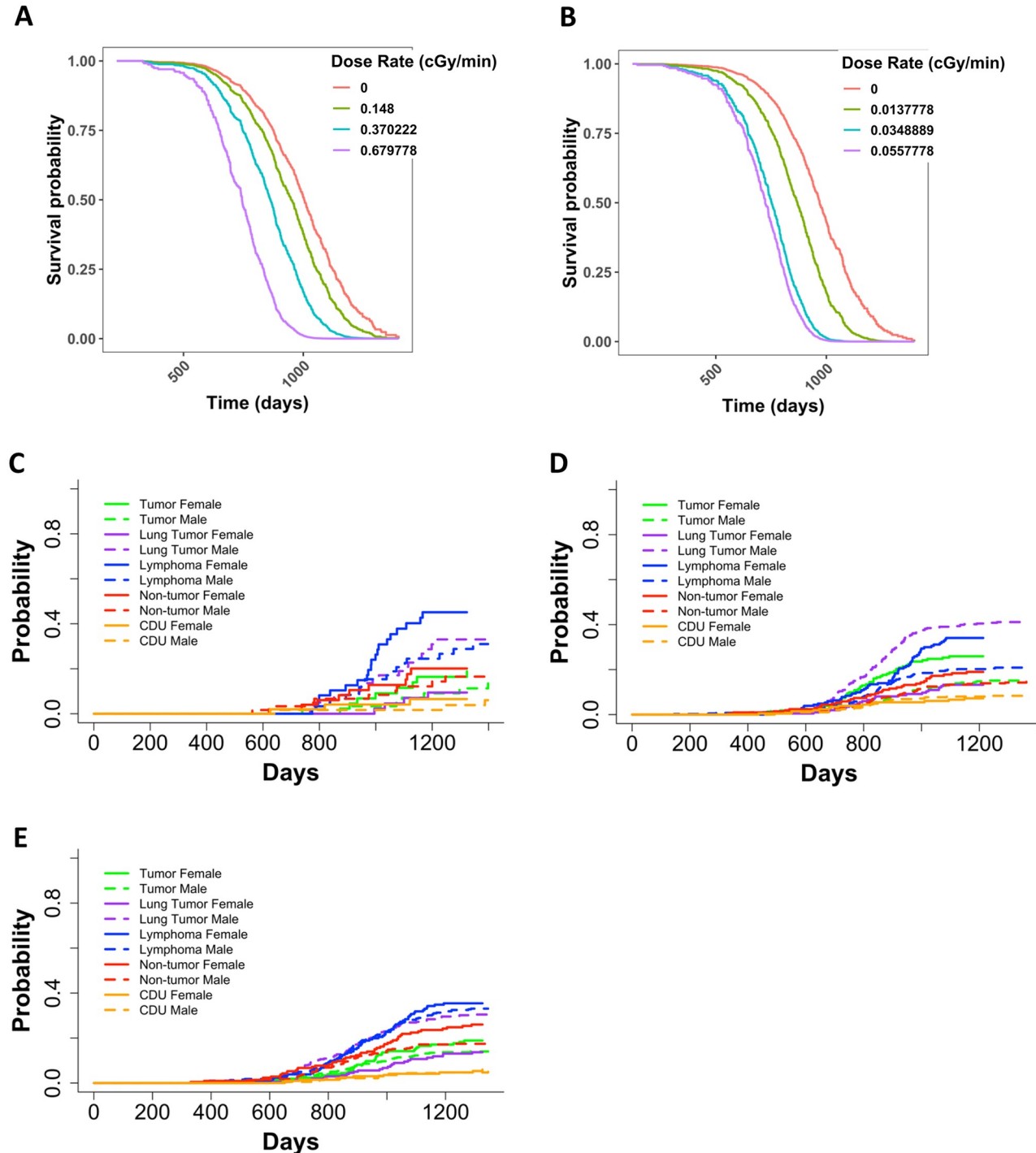

**Fig 5.** Representative graphs from Cox PH model with age at death as the time scale and sex and categorical dose rate as independent variables for **(A)** JM8 gamma irradiated mice and **(B)** JM8 neutron irradiated mice. The predicted outcomes shown in **A** and **B** represent female mice. Non-parametric CIFs for the five main categories of COD, grouped by sex for JM8 mice that were **(C)** sham irradiated, **(D)** neutron irradiated, or **(E)** gamma irradiated. P-values for differences between sexes in **(C)**: tumor = 0.46, lung tumor = 0.0073, lymphoma = 0.085, non-tumor = 0.62, CDU = 0.51, **(D)**: tumor = 7.37E-04, lung tumor = 6.24E-11, lymphoma = 1.04, non-tumor = 0.139, CDU = 0.938 and **(E)**: tumor = 0.18, lung tumor = 5.13E-06, lymphoma = 0.68, non-tumor = 0.029, CDU = 0.57.

atomic bomb survivors who were exposed acutely and predominantly to gamma rays, the excess relative risk decreased approximately 21% per decade of age first irradiated [41]. In general, it has been accepted that exposures later in life are less risky because the latent phase for slower developing diseases, such as tumors, extends past the average lifespan. In this study, the comparisons between mice irradiated at a younger age vs. 500 days old showed a shift in type of disease as a cause of death. For both radiation qualities, accumulation of solid tumors as a cause of death was the highest in aged animals and this trend was more pronounced in the neutron irradiated animals. This was different from the prevalent cause of death in control animal or animals irradiated at a younger age where lymphoma was the most frequent cause of death. The most striking difference between gamma and neutron irradiated mice is that the average lifespan of the neutron irradiated animals increases with the later age at exposure. Because high LET ionizing radiation is more likely to produce clustered DNA damage, which is more difficult to repair [1] and because aged mice are less likely to have efficient cellular recovery mechanisms, it is possible that neutron irradiation late in life may lead to permanent senescence of latent cancer stem cells. Similar hypotheses were stated by others [42, 43] but never in conjunction with neutron exposures. Recent studies investigating radiation sensitivity for different age at exposure have shown that ionizing radiation could may put younger populations at increased risk due to disease initiation, while in substantially older groups have an increased cancer risk may be associated with increased cancer progression, with an overall bimodal risk distribution with respect to age [44]. The large increase in tumor death incidences we observed in neutron irradiated aged mice agrees with this finding, while at the same time risk of lymphoma becomes lower and the average age of animals increases for the same total dose. Further work will be necessary to explore these novel findings.

Fractionation decreased the death hazard in neutron irradiated mice (Fig 1D and Table 1). Some effects of fractionation on neutron exposures were noted previously [9, 17, 20, 28, 45], albeit with mixed conclusions, possibly related to the time between fractions. For example, in female BALB/c mice lung adenocarcinoma incidence was increased by as much as two-fold when animals where exposed to fractionated fission neutrons with 24 h between fractions [46]. On the other hand, a study by De Majo [31] that included a one-day-five-fractions fractionation schedule, found no significant differences for lung cancer incidence in BC3F1 mice. In a study examining lung cancer risk using the Janus dataset, Heidenreich et al. found that fractionation slightly reduced the relative risk of mice exposed to less than 0.3Gy, but had a detrimental effect at higher total doses [47]. We found that fractionation was only protective for tumor deaths and did not play a significant role for any other COD group we analyzed (Fig 2, Table 2, **and S4 Fig in** S1 File). Exploring the mechanism for decreased death hazard through fractionation for mice that died with tumors could lead to potential new radiation treatment or radiation protection strategies.

Lung tumors showed distinct differences in incidence based on sex and response to fractionation. While fractionation was protective against tumor deaths in mice exposed either to neutrons or gamma rays, fractionation was not protective for lung tumors in either case [14]. In humans, females exposed to ionizing radiation are at a greater risk for lung tumors compared to males [45, 46]. The differences in etiology between lung tumors and all other tumors are worthy of further investigation.

In gamma and neutron irradiated mice, the hazard of death was greater in females compared to males for all causes of death, except for lung tumors. This was one of our most robust findings with cause-specific hazards, subdistribution hazards, and crude CIF results all in agreement for both qualities of radiation. The increased risk for lung tumors in males compared to females, however, may be specific to B6CF1 mice. Reports on lung tumor incidence rates in RFM mice suggested that males exposed to 10 Gy of x-rays were more resistant to lung

tumor development than females exposed to 9 Gy of x-rays, the opposite from the trend seen in B6CF1 mice [29, 48, 49]. The breathing rate of mice after radiation exposures could potentially contribute to differences in lung tumor susceptibility between sexes. It is known that breathing rates increase after high dose radiation exposures, creating a high oxidative state with more free radicals that many be causing tumor initiation and progression [50, 51]. Studies examining differences in breathing rates between male and female mice have yet to be conducted, but they could lead to new insights in lung tumor development differences between sexes. Tissue samples from mice in the Janus Tissue Archive are available for conducting more detailed molecular biology experiments, such as the PCR based study carried out on Rb and p53 gene deletions in lung adenocarcinomas [52]. It is also possible that there is a contribution of genetics to this difference in response since RFM and B6CF1 mice have many genetic differences, as one would predict for different mouse strains.

Exploring the differences in disease patterns between aged mice exposed to neutrons or gamma rays led to several significant findings (Fig 4). First, we discovered that lymphoma is the most frequent COD in control mice and gamma irradiated mice at a younger age, but that tumors are equally frequent COD as lymphoma in neutron irradiated mice. This distinction could lead to new hypotheses regarding the differences in disease initiation and progression between lymphomas and solid tumors. Second, in aged gamma irradiated mice the frequency of tumors "catches up" with the lymphomas while in neutron irradiated mice tumors become predominant COD. Finally, the non-parametric CIF curves also showed that the age of death was earlier in control and gamma irradiated mice compared to aged neutron irradiated mice (Fig 4). The mechanisms responsible for these differences have yet to be revealed.

Finally, this work included neutron and gamma irradiated mice that were irradiated once weekly until death, making our intention-to-treat analysis method non suitable [53]. Instead, we analyzed the data on these animals using dose rate as the independent variable. Not surprisingly, we found a significant increase in the death hazard as dose rate increased both in gamma and neutron irradiated mice (Fig 5A and 5B). Gamma ray dose rate increase resulted in gradual decrease in the survival probability for dose rates up to 0.048cGy, but higher dose-rates of 0.37cGy/min to 0.68cGy/min had a greater health impact. Conversely, among neutron irradiated mice, detriment from weekly irradiation was marked with dose rate increase until it nearly plateaued with little difference in the survival curves plotted for mice exposed to weekly neutron doses of 1.58 or 2.52 cGy.

In conclusion, whole body exposures to fission spectrum neutrons evaluated in this work suggest that factors such as age at exposure and fractionation play a decisive role in the ultimate health outcomes from neutron exposures. Because these findings suggest that the passage of time increases resilience to neutron exposure we should investigate the role of biological mechanisms engaged in neutron irradiated whole organisms in our efforts to apprehend the effects of neutron exposures.

## Supporting information

**S1 File.**
(PDF)

## Acknowledgments

The authors of this paper publicly acknowledge the support of Edward Malthouse and Benjamin Haley for their support and statistical expertise. We also thank Carissa Ritner for her support and thoughtful conversations.

## Author Contributions

**Conceptualization:** Alia Zander, Tatjana Paunesku, Gayle E. Woloschak.

**Formal analysis:** Alia Zander.

**Funding acquisition:** Tatjana Paunesku, Gayle E. Woloschak.

**Investigation:** Alia Zander, Tatjana Paunesku.

**Methodology:** Alia Zander, Tatjana Paunesku, Gayle E. Woloschak.

**Project administration:** Alia Zander, Tatjana Paunesku, Gayle E. Woloschak.

**Resources:** Alia Zander, Tatjana Paunesku, Gayle E. Woloschak.

**Software:** Alia Zander.

**Supervision:** Tatjana Paunesku, Gayle E. Woloschak.

**Validation:** Alia Zander.

**Visualization:** Alia Zander.

**Writing – original draft:** Alia Zander.

**Writing – review & editing:** Alia Zander, Tatjana Paunesku, Gayle E. Woloschak.

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
