## [Decision Letter · Decision Letter 0]

25 Sep 2020

PONE-D-20-08070

Analyses of cancer incidence and other morbidities in neutron irradiated B6CF1 mice

PLOS ONE

Dear Dr. Woloschak,

Thank you for submitting your manuscript to PLOS ONE. After careful consideration, we feel that it has merit but does not fully meet PLOS ONE’s publication criteria as it currently stands. Therefore, we invite you to submit a revised version of the manuscript that addresses the points raised during the review process.

We look forward to receiving your revised manuscript.

Kind regards,

Norman J Kleiman, PhD

Academic Editor

PLOS ONE

Additional Editor Comments:

Thank you for your submission. Both reviewers offer helpful counsel and suggestions to improve this manuscript. In particular, one suggests a somewhat different statistical approach which might prove more helpful in addressing some of the concerns. Please review their comments carefully. We look forward to your response and resubmission.

Journal Requirements:

"The authors have declared that no competing interests exist.".

We note that the CA (Gayle E. Woloschak) is a section editor for PLOS ONE.

i) Please confirm that this does not alter your adherence to all PLOS ONE policies on sharing data and materials, by including the following statement: "This does not alter our adherence to  PLOS ONE policies on sharing data and materials.” (as detailed online in our guide for authors http://journals.plos.org/plosone/s/competing-interests).  If there are restrictions on sharing of data and/or materials, please state these. Please note that we cannot proceed with consideration of your article until this information has been declared.

ii) Please include your updated Competing Interests statement in your cover letter; we will change the online submission form on your behalf.

3. Please include a caption for figure 4.

Reviewers' comments:

Reviewer's Responses to Questions

**Comments to the Author**

1. Is the manuscript technically sound, and do the data support the conclusions?

Reviewer #1: Yes

Reviewer #2: Yes

2. Has the statistical analysis been performed appropriately and rigorously? 

Reviewer #1: Yes

Reviewer #2: Yes

3. Have the authors made all data underlying the findings in their manuscript fully available?

Reviewer #1: Yes

Reviewer #2: Yes

4. Is the manuscript presented in an intelligible fashion and written in standard English?

Reviewer #1: Yes

Reviewer #2: Yes

5. Review Comments to the Author

Reviewer #1: This manuscript reports a new analysis of a series of large-scale neutron irradiation experiments held at

Argonne National Laboratory between 1972 and 1989. These experiments produced a dataset on over 50,000 male and female mice which is a mine of important discoveries still underutilized.

Using a new approach to utilize experimental controls to determine whether a cross comparison between experiments was appropriate, the Authors amalgamated data on neutron exposures and reached important conclusions.

The first one is that dose fractionation significantly improved overall survival having a significant impact on the death hazard for animals that died of solid tumors, but did not significantly impact any other causes of death.

The second one regarded the dominant cause of death that was represented by solid tumors in neutron irradiated mice while lymphomas were the dominant cause of death in gamma irradiated mice.

Additionally, the Authors propose that irradiation at advanced age played a significant role in decreasing the death hazard for neutron irradiated mice, but not for gamma irradiated mice. Mice that were 500 days old before their first exposures to neutrons began dying later than both sham irradiated or gamma irradiated mice.

These conclusions are very well supported by the statistical analyses and of great interest not only for radiobiologists but also for a more general audience and the work certainly deserves publication on PlosOne. On the other hand, there are some presentation defects and inaccuracies that need to be corrected in order to improve the manuscript before publication (see below).

Major points

- Figure 1 panel C does not correspond to the legend

C) Age at death in days versus total dose in Gy. Colors indicate whether a mouse was first irradiated before or after 500 days.

- Legend to Fig. 1. All independent variables were significant in the model, as shown by the parameter estimates and statistical output from the model (E).

I do not see panel E

- Line 234: The interaction was beneficial for survival in groups of animals that developed tumors and lymphoma (Fig 2B and Table 2) and detrimental for mice dying of non tumors and CDU (Fig 2C and 2D and Table 2).

Here I have difficulties in interpreting the figures. Why the acute curve does not show in panel B and C at 2 Gy? If this is because there is no survival at this acute dose, then fractionation is beneficial also for mice dying for non-tumors. Fractionation looks detrimental to me only for unknown cause of death (2D).

- Legend to Figure 4 is missing so it is difficult to follow the text.

- Line 333: Control mice (Fig 4A) and gamma irradiated mice (Fig 4 B-D) all displayed similar trends in COD patterns with lymphomas as the most prevalent, followed by tumors, non-tumors, and CDU.

In panel 4D Lymphomas does not seem the most prevalent.

Discussion

- Line 406: However, the stark contrast in effect for age first irradiated between gamma and neutron irradiated mice indicates that a longer time for disease development is not completely sufficient for increasing risk in younger mice. Because high LET ionizing radiation is more likely to produce clustered DNA damage, which is more difficult to repair (1) and because aged mice are less likely to have efficient cellular recovery mechanisms, it is possible that neutron irradiation late in life may lead to permanent senescence of latent cancer stem cells.

Here the Authors should also take in account that aged mice exposed to neutrons die significantly more of tumors and non-tumors, and significantly less of lymphomas which is discussed below but could be also relevant in this issue.

- Line 434: In gamma and neutron irradiated mice, the hazard of death was greater in males

compared to females for all causes of death, except for lung tumors.

Should’nt be the other way around?

- Line 457: Second, aged mice exposed to gamma rays died significantly more of non-tumors and died significantly less of lymphomas compared to neutron irradiation survivors ?????.

Neutron irradiation survivors do not die…. The phrase is cryptic

Minor points

Line 199 should read: life shortening per cumulative dose was estimated to be around 10 for neutron doses up to 40cGy and....

Line 267 should read: 5 main categories of COD grouped by sex for control/sham irradiated mice.....

There are some other typos and a careful text revision should be done

Reviewer #2: The manuscript “Analyses of cancer incidence and other morbidities in neutron irradiated B6CF1 mice” used a very large data set, generated by pooling various experimental results on neutron and gamma ray mouse irradiations from the JANUS archive. Cox proportional hazard models were used to analyze the data and quantify the effects of neutron dose, dose fractionation, and other variables. The paper reports some important results, but its methods would be improved by addressing the following issues:

The authors “used Cox proportional hazards models with sex and a categorical fractionation term as independent variables for the main model”.

If I understand correctly, only fixed effects were used in the models. However, random effects would be very likely be helpful in the model because mouse responses could be correlated by experiment type, year, etc. The authors could use Mixed Effects Cox Models (coxme R package) to address the potential effects of such factors.

The authors used separate terms for total dose, dose fractionation (categorical), and an interaction of these. The resulting parameter estimates are sometimes not very intuitive, e.g. positive for fractionated and negative for interaction in Tables 1-2. A more mechanistically plausible explanation could be provided by using the Biologically Effective Dose (BED) concept (e.g. 21 years of biologically effective dose, JF Fowler - The British journal of radiology, 2010). It conveniently integrates total dose and fractionation, with alpha/beta being set to some plausible values (e.g. 10 Gy). BED could be inserted into the Cox model instead of separate terms for total dose and fractionation, reducing parameter number by 1.

Was there any testing for multicollinearity in the Cox models?

The authors state “In Survival Analysis for JM8 mice, dose rates are treated as categorical variables.” Why not try some continuous response to dose rate/dose which is biologically meaningful?

The authors state “Because the greatest number of fractions a neutron irradiated mouse received was only 60 fractions, we also excluded all animals with more than 60 sham irradiation fractions from this work.” Would it make a difference not to exclude them?

The authors state “we used Kaplan Meier survival curves to validate the proportional hazards assumption in our model (S1 Fig E)”. It is not clear from this figure panel how the validation was done.

What are “nuanced variables” that the authors refer to?

The authors state “lung tumors were more common after fractionated exposures”. Perhaps this represents an inverse dose rate / protraction effect, which is known for lung cancers induced by prolonged high-LET radiation. Could be mentioned in the discussion.

Lines 403-404: formatting issue

6. PLOS authors have the option to publish the peer review history of their article (what does this mean?). If published, this will include your full peer review and any attached files.

Reviewer #1: No

Reviewer #2: No

---

## [Author Response · Author response to Decision Letter 0]

27 Nov 2020

Dear Dr Kleiman-

We have made numerous changes on our manuscript PONE-D-20-08070 “Analyses of cancer incidence and other morbidities in neutron irradiated B6CF1 mice” as suggested by you and the reviewers. We are grateful for your comments and believe that the manuscript is now significantly improved. In the text below – comments from the reviewers will be in italics, our responses in regular front and the excerpts of the newly added text are underlined.

Editor’s comments requested that we update the cover letter by adding the information on GEW’s editor status.

We have made this addition and now attach a new version of the cover letter.

Editor’s comments: Please include a caption for figure 4

We have used a text box for this caption in error (and the box became “hidden” between two pages). We now deleted this text box and added Figure 4 caption in the body of the document:

“Figure 4: Non-parametric CIFs for the four main categories for COD in (A) control mice, (B) gamma irradiated mice first irradiated under 500 days old, (C) gamma irradiated mice that were first irradiated under 500 days old with a dose cut-off of 6Gy, (D) gamma irradiated mice first irradiated over 500 days old, (E) neutron irradiated mice first irradiated under 500 days old, and (F) neutron irradiated mice first irradiated over 500 days old. P-values for differences between groups of ages first irradiated in C and D: tumor = 0.73, lymphoma = 7.03E-05, non-tumor = 9.73E-13, CDU = 0.09, and in E and F: tumor = 2.30E-03, lymphoma = 3.71E-08, non-tumor = 2.49E-04, and CDU = 0.97.”

We have noticed that we used text boxes for figure legends in several other instances. In order to preempt repetition of the problem with Figure 4 caption we have now-redone these captions as regular text. Also, in the process of correcting this figure legend we found that we were not consistent with placing later labels. Consequently, we made adjustments of latter placement in the captions as necessary in order to decrease possible confusion. 

Review Comments to the Authors: Reviewer 1

Reviewer #1: This manuscript reports a new analysis of a series of large-scale neutron irradiation experiments held at Argonne National Laboratory between 1972 and 1989. These experiments produced a dataset on over 50,000 male and female mice which is a mine of important discoveries still underutilized. Using a new approach to utilize experimental controls to determine whether a cross comparison between experiments was appropriate, the Authors amalgamated data on neutron exposures and reached important conclusions. 

The first one is that dose fractionation significantly improved overall survival having a significant impact on the death hazard for animals that died of solid tumors, but did not significantly impact any other causes of death.

The second one regarded the dominant cause of death that was represented by solid tumors in neutron irradiated mice while lymphomas were the dominant cause of death in gamma irradiated mice.

Additionally, the Authors propose that irradiation at advanced age played a significant role in decreasing the death hazard for neutron irradiated mice, but not for gamma irradiated mice. Mice that were 500 days old before their first exposures to neutrons began dying later than both sham irradiated or gamma irradiated mice.

These conclusions are very well supported by the statistical analyses and of great interest not only for radiobiologists but also for a more general audience and the work certainly deserves publication on PlosOne. On the other hand, there are some presentation defects and inaccuracies that need to be corrected in order to improve the manuscript before publication (see below).

We are very grateful for your nice words and detailed comments!

Figure 1 panel C does not correspond to the legend C) Age at death in days versus total dose in Gy. Colors indicate whether a mouse was first irradiated before or after 500 days.

Thank you for noticing this issue (we have swapped the panels by accident). The legend is now corrected.

“Fig 1: Analysis on animals filtered in S1 Table that were controls or neutron irradiated. (A) Box plot of age at death in days versus total dose in Gy. Colors indicate if the exposure was acute or fractionated. (B) Age at death in days versus total dose in Gy. Colors indicate whether a mouse was first irradiated before or after 500 days. (C) Histogram of the total number of animals versus age first irradiated in days….”

Legend to Fig. 1. All independent variables were significant in the model, as shown by the parameter estimates and statistical output from the model (E). 

I do not see panel E

The text should have referred to Table 1, this error is now fixed. “All independent variables were significant in the model, as shown by the parameter estimates and statistical output from the model (Table 1).”

Line 234: The interaction was beneficial for survival in groups of animals that developed tumors and lymphoma (Fig 2B and Table 2) and detrimental for mice dying of non tumors and CDU (Fig 2C and 2D and Table 2).

Here I have difficulties in interpreting the figures. Why the acute curve does not show in panel B and C at 2 Gy? If this is because there is no survival at this acute dose, then fractionation is beneficial also for mice dying for non-tumors. Fractionation looks detrimental to me only for unknown cause of death (2D).

We have now tried to fix the confusing text and we also add that in Figure 2 for 2 Gy in panels B and C the acute line is directly behind the fractionation line.

“Accounting for competing risks by calculating cause-specific hazards, we found that the interaction between total dose and the fractionation variable was significant for tumor deaths (Fig 2A and Table 2) where this interaction was beneficial for survival. In animals that developed lymphoma (Fig 2B and Table 2), non-tumors (Fig 2C and Table 2) or CDU (Fig 2B and Table 2) fractionation lead to differences that were not significant. In figures 2B and 2C the line representing acute exposures is hidden directly behind the line showing fractionated exposures.”

Legend to Figure 4 is missing so it is difficult to follow the text.

As we have added in our responses to the editor 

“Figure 4: Non-parametric CIFs for the four main categories for COD in (A) control mice, (B) gamma irradiated mice first irradiated under 500 days old, (C) gamma irradiated mice that were first irradiated under 500 days old with a dose cut-off of 6Gy, (D) gamma irradiated mice first irradiated over 500 days old, (E) neutron irradiated mice first irradiated under 500 days old, and (F) neutron irradiated mice first irradiated over 500 days old. P-values for differences between groups of ages first irradiated in C and D: tumor = 0.73, lymphoma = 7.03E-05, non-tumor = 9.73E-13, CDU = 0.09, and in E and F: tumor = 2.30E-03, lymphoma = 3.71E-08, non-tumor = 2.49E-04, and CDU = 0.97.”

Line 333: Control mice (Fig 4A) and gamma irradiated mice (Fig 4 B-D) all displayed similar trends in COD patterns with lymphomas as the most prevalent, followed by tumors, non-tumors, and CDU.

In panel 4D Lymphomas does not seem the most prevalent.

Thank you for your comment - the portion of a paragraph below was edited with changes underlined. With the added caption for figure 4 we hope that this will now make more sense.

“Because of the pronounced difference in longevity of aged mice exposed to neutrons, we examined causes of death in mice irradiated when aged versus those that were irradiated younger. In addition, we compared them with the mice exposed to gamma rays at 500 days of age. Control mice (Fig 4A) and gamma irradiated mice first irradiated under 500 days old (Figs 4B and 4C) all displayed similar trends in COD patterns with lymphomas as the most prevalent, followed by tumors, non-tumors, and CDU.”

Discussion

Line 406: However, the stark contrast in effect for age first irradiated between gamma and neutron irradiated mice indicates that a longer time for disease development is not completely sufficient for increasing risk in younger mice. Because high LET ionizing radiation is more likely to produce clustered DNA damage, which is more difficult to repair (1) and because aged mice are less likely to have efficient cellular recovery mechanisms, it is possible that neutron irradiation late in life may lead to permanent senescence of latent cancer stem cells.

Here the Authors should also take in account that aged mice exposed to neutrons die significantly more of tumors and non-tumors, and significantly less of lymphomas which is discussed below but could be also relevant in this issue.

Thank you for this suggestion – we have added a new sentence to this section:

“In this study, the comparisons between animals irradiated at a younger age vs. aged showed a shift in type of disease as a cause of death. For both radiation qualities, accumulation of solid tumors as a cause of death was the highest in aged animals and this was more pronounced in the neutron irradiated animals. This was different from the prevalent cause of death in control animal or animals irradiated at a younger age where lymphoma was the most frequent cause of death.” 

Line 434: In gamma and neutron irradiated mice, the hazard of death was greater in males compared to females for all causes of death, except for lung tumors.

Should’nt be the other way around?

Yes, thank you, this is now fixed. The corrected sentence is “In gamma and neutron irradiated mice, the hazard of death was greater in females compared to males for all causes of death, except for lung tumors.”

Line 457: Second, aged mice exposed to gamma rays died significantly more of non-tumors and died significantly less of lymphomas compared to neutron irradiation survivors ?????.

Neutron irradiation survivors do not die…. The phrase is cryptic

Yes, thank you, the text was confusing and we have removed some text and added some new text:

“Exploring the differences in disease patterns between aged mice exposed to neutrons or gamma rays led to several significant findings (Fig 4). First, we discovered that lymphoma is the most frequent COD in control mice and gamma irradiated mice at a younger age, but that tumors are equally frequent COD as lymphoma in neutron irradiated mice. This distinction could lead to new hypotheses regarding the differences in disease initiation and development between lymphomas and solid tumors. Second, in aged gamma irradiated mice the frequency of tumors “catches up” with the lymphomas while in neutron irradiated mice tumors become clearly predominant COD. Finally, the non-parametric CIF curves also showed that the age of death was earlier in control and gamma irradiated mice compared to aged neutron irradiated mice (Fig 4). The mechanisms responsible for these differences have yet to be revealed.”

Minor points

Line 199 should read: life shortening per cumulative dose was estimated to be around 10 for neutron doses up to 40cGy and…

Word was added:

“… to be around 10 for neutron doses up to 40cGy and approximately 5 or less for neutron doses over 40cGy …”

Line 267 should read: 5 main categories of COD grouped by sex for control/sham irradiated mice.....

Correction was made.

There are some other typos and a careful text revision should be done

Thank you, we found a few other typos as well as some labeling inconsistencies. These changes should be clear from the track changes version of the document.

Reviewer 2:

The manuscript “Analyses of cancer incidence and other morbidities in neutron irradiated B6CF1 mice” used a very large data set, generated by pooling various experimental results on neutron and gamma ray mouse irradiations from the JANUS archive. Cox proportional hazard models were used to analyze the data and quantify the effects of neutron dose, dose fractionation, and other variables. The paper reports some important results, but its methods would be improved by addressing the following issues:

The authors “used Cox proportional hazards models with sex and a categorical fractionation term as independent variables for the main model”. If I understand correctly, only fixed effects were used in the models. However, random effects would be very likely be helpful in the model because mouse responses could be correlated by experiment type, year, etc. The authors could use Mixed Effects Cox Models (coxme R package) to address the potential effects of such factors.

We are grateful for this suggestion and will look into using coxme package more extensively as we approach this dataset again. At this point of time, however, complete redoing of this manuscript would make it difficult for a comprehensive joint consideration of this study with the work that we have already done with the gamma irradiated mice (reference 14: Zander A, Paunesku T, Woloschak GE. Analyses of cancer incidence and other morbidities in gamma irradiated B6CF1 mice. PloS one. 2020;15(8):e0231510.) where the main body of our work with control mice dataset was published.

Nevertheless, we did use the coxme R package to generate an additional supplemental table (S2 Table) that supports our work in this study as well as the one that was already published (14) and we discuss this in the methods section of the manuscript as follows:

“We used Cox proportional hazards models with sex and a categorical fractionation term as independent variables for the main model (S1 Fig A and B). For sensitivity analysis, we also stratified by sex (S 1 Fig C and D) and found no significant changes in the model output. In addition, we tested the impact of using a mixed-effects model to account for variability between experiments. The results in S2 Table show similar results to the main model. Model output showed that the integrated and penalized log likelihoods were very similar and both significant (p-value < 0.001), increasing our confidence that the control experiments outlined in our previous paper (14) were sufficient for capturing variability between experiments.”

S2 Table is now also included:

Term Estimate Hazard Ratio P-value

Sex(M) -0.210 0.810 (0.77,0.85) <0.001

Fractionated 0.023 1.023 (0.88, 1.18) 0.76

Total Dose 0.019 1.019 (1.01, 1.02) <0.001

Age First Irradiated -0.002 0.998 (0.99, 1.00) 0.23

Fractionated:Total Dose -0.010 0.990 (0.99, 0.99) <0.001

S2 Table. Results from mixed effects model using the coxme function from the coxme R package. The fixed effects covariates were sex, fractionation, total dose, an interaction term between the total dose and fractions, and age first irradiated. The random effect covariate was experiment number as a categorical variable.

The authors used separate terms for total dose, dose fractionation (categorical), and an interaction of these. The resulting parameter estimates are sometimes not very intuitive, e.g. positive for fractionated and negative for interaction in Tables 1-2. A more mechanistically plausible explanation could be provided by using the Biologically Effective Dose (BED) concept (e.g. 21 years of biologically effective dose, JF Fowler - The British journal of radiology, 2010). It conveniently integrates total dose and fractionation, with alpha/beta being set to some plausible values (e.g. 10 Gy). BED could be inserted into the Cox model instead of separate terms for total dose and fractionation, reducing parameter number by 1.

Thank you for highlighting the possibility of using BED for our work. We chose not to use BED because it implies alpha-beta ratios and linear quadratic formula vs. tumor control in therapy. We appreciate the value of BED for therapy as explained in 21 years of Biologically Effective Dose, JF Fowler 2010, but in the current context where we are studying tumor initiation and promotion (for which our mice give indirect evidence) use of BED would require an overhaul of this concept that is beyond the scope of this work.

Was there any testing for multicollinearity in the Cox models?

Yes, we used the vif function from the car package to check for multicollinearity. We made this more explicit by adding the following text to the methods section. 

“We checked for multicollinearity in our models using the vif function from the car package in R. As expected, there was multicollinearity with the interaction term, however it was not an issue in our model because the standard error in our models was not overly inflated. Additionally, sensitivity analyses show consistent results across different models.”

The authors state “In Survival Analysis for JM8 mice, dose rates are treated as categorical variables.” Why not try some continuous response to dose rate/dose which is biologically meaningful?

The dataset JM8 comes from an experiment where the dose rates and dose per fraction were fixed and the mice were treated once a week until death and therefore do not satisfy the conditions that would have allowed it to be included in the main analyses (e.g. please note the manner in which JM8 was mentioned in S2 Table “Experimental design – separate analysis required”). Therefore, the mice that lived longest received higher doses within each dose rate study group. 

We now add the following text to the Methods section in order to make this experimental setup more clear

“One of the Janus experiments excluded from the main analysis is the experiment JM8 where animals received once a week a 45 minutes sham fraction or a fraction of gamma rays (dose rates of 0.15, 0.37, or 0.68 cGy/min), or neutrons (dose rates 0.014, 0.035 or 0.056 cGy/min). Because irradiations continued until the death of the animals, it was impossible to evaluate discrete total doses in this study.”

The authors state “Because the greatest number of fractions a neutron irradiated mouse received was only 60 fractions, we also excluded all animals with more than 60 sham irradiation fractions from this work.” Would it make a difference not to exclude them?

In our companion publication (reference 14: Zander A, Paunesku T, Woloschak GE. Analyses of cancer incidence and other morbidities in gamma irradiated B6CF1 mice. PloS one. 2020;15(8):e0231510.) we give extensive analyses of the control mice dataset. To recapitulate – we have found that 5 fractions per week, 60 weeks sham irradiations (300 sham fractions in all) caused stress to animals that causing them to die significantly earlier than other control mice, and excluded all 300 fractions work from the gamma radiation paper (14). 

In order to be conservative while working on this paper, we limited our control dataset to the same maximum number of sham fractions as was received by the neutron irradiated mice. 

The authors state “we used Kaplan Meier survival curves to validate the proportional hazards assumption in our model (S1 Fig E)”. It is not clear from this figure panel how the validation was done.

We updated this section to try to clarify this issue. 

“Finally, we used Kaplan Meier survival curves to validate the proportional hazards assumption in our model and found non-overlapping survival curves between mice that received acute and fractionated exposures.”

What are “nuanced variables” that the authors refer to?

We are referring to variables that do not have proportional hazards. Thank you for pointing out this vague description, we updated the text:

“Cox PH models enabled us to include interactions between variables, incorporate quantitative and categorical variables, and stratify by variables that do not have proportional hazards (34). “

The authors state “lung tumors were more common after fractionated exposures”. Perhaps this represents an inverse dose rate / protraction effect, which is known for lung cancers induced by prolonged high-LET radiation.

Thank you for this suggestion. To emphasize different findings in this field of research we have added more to the discussion as follows:

“Fractionation decreased the death hazard in neutron irradiated mice (Fig 1D and Table 1). Some effects of fractionation on neutron exposures were noted previously (9, 17, 20, 28, 45), albeit with mixed conclusions, possibly related to the time between fractions. For example, in female BALB/c mice lung adenocarcinoma incidence was increased by as much as two-fold when animals where exposed to fractionated fission neutrons with 24 h between fractions (46). On the other hand, a study by De Majo (31) that included a one-day-5 fractions fractionation schedule, found no significant differences were found for lung cancer incidence in BC3F1 mice. In a study examining lung cancer risk using the Janus dataset, Heidenreich et al. found that fractionation slightly reduced the relative risk of mice exposed to less than 0.3Gy, but had a detrimental effect at higher total doses (47).”

Lines 403-404: formatting issue

We have fixed this issue, thank you.

At this time, we hope that the manuscript is completely corrected and that it will meet the expectations of the reviewers.

Sincerely-

Gayle 

Gayle E Woloschak, Ph. D., FASTRO, Professor

Departments of Radiation Oncology, Radiology, and Cell and Molecular Biology

Robert H. Lurie Comprehensive Cancer Center

Feinberg School of Medicine, Northwestern University

300 E. Superior St., Tarry 4-760

Chicago, IL 60611

312-503-4322 

g-woloschak@northwestern.edu

---

## [Decision Letter · Decision Letter 1]

3 Feb 2021

Analyses of cancer incidence and other morbidities in neutron irradiated B6CF1 mice

PONE-D-20-08070R1

Dear Dr. Woloschak,

We’re pleased to inform you that your manuscript has been judged scientifically suitable for publication and will be formally accepted for publication once it meets all outstanding technical requirements.

Kind regards,

Nobuyuki Hamada

Academic Editor

PLOS ONE

Additional Editor Comments (optional):

Reviewers' comments:

Reviewer's Responses to Questions

**Comments to the Author**

1. If the authors have adequately addressed your comments raised in a previous round of review and you feel that this manuscript is now acceptable for publication, you may indicate that here to bypass the “Comments to the Author” section, enter your conflict of interest statement in the “Confidential to Editor” section, and submit your "Accept" recommendation.

Reviewer #1: All comments have been addressed

Reviewer #2: All comments have been addressed

2. Is the manuscript technically sound, and do the data support the conclusions?

Reviewer #1: Yes

Reviewer #2: Yes

3. Has the statistical analysis been performed appropriately and rigorously? 

Reviewer #1: Yes

Reviewer #2: Yes

4. Have the authors made all data underlying the findings in their manuscript fully available?

Reviewer #1: Yes

Reviewer #2: Yes

5. Is the manuscript presented in an intelligible fashion and written in standard English?

Reviewer #1: Yes

Reviewer #2: Yes

6. Review Comments to the Author

Reviewer #1: (No Response)

Reviewer #2: The authors adequately and appropriately addressed all my questions from the previous review. The paper is now good for publication.

7. PLOS authors have the option to publish the peer review history of their article (what does this mean?). If published, this will include your full peer review and any attached files.

Reviewer #1: **Yes: **Rodolfo Negri

Reviewer #2: No

---

## [Editor Report · Acceptance letter]

9 Feb 2021

PONE-D-20-08070R1 

Analyses of cancer incidence and other morbidities in neutron irradiated B6CF1 mice 

Dear Dr. Woloschak:

I'm pleased to inform you that your manuscript has been deemed suitable for publication in PLOS ONE. Congratulations! Your manuscript is now with our production department. 

Kind regards, 

on behalf of

Dr. Nobuyuki Hamada 

Academic Editor

PLOS ONE